# An Adaptive Weight Model Predictive Control Algorithm to Trajectory Tracking Control of UUV

Danjie Zhu
*School of Naval Architecture and Ocean Engineering*
*Jiangsu University of Science and Technology*
Zhenjiang, China
zhudans13@163.com

Hongtan Zhao
*Laboratory of Underwater Vehicles and Intelligent Systems*
*Shanghai Maritime University*
Shanghai, China
zhaohongtan@foxmail.com

Bing Sun*
*Laboratory of Underwater Vehicles and Intelligent Systems*
*Shanghai Maritime University*
Shanghai, China
bingsun@shmtu.edu.cn

Zinan Su
*Laboratory of Underwater Vehicles and Intelligent Systems*
*Shanghai Maritime University*
Shanghai, China
suzn_frank@outlook.com

*Abstract*—**Unmanned Underwater Vehicle (UUV) has been applied increasingly in marine work, and the trajectory tracking speed and accuracy directly affect work efficiency. To improve the convergence speed and accuracy of UUV tracking control, an adaptive weight model predictive control based on quantum particle swarm optimization (QPSO-AWMPC) is presented in this paper. The control weight is determined by the state tracking error. At the initial stage, where the tracking error is large, the weight control is lowered to achieve a higher speed. At the stable phase, where the tracking error is small, the weight control is raised to maintain robustness. A kinematics controller is designed to optimize state error and obtain the desired speed within constraints, while a dynamic controller is proposed to optimize speed and obtain the required thrust. Finally, the three-dimensional simulation verifies the effectiveness of the proposed method. Compared with the traditional MPC and Backstepping methods, the internal constraint of the UUV is satisfied while the convergence speed, accuracy, and robustness are improved.**

*Index Terms*—**Trajectory tracking, Model predictive control, Convergence speed, Adaptive weight**

## I. INTRODUCTION

Unmanned Underwater Vehicle (UUV), due to the flexibility and autonomy, is used to explore the oceanic and scientific investigation. Trajectory tracking is an important aspect of the control research of UUV. Now many trajectory tracking methods have been proposed, such as PID control [1, 2], backstepping control [3, 4], neural network method [5], sliding mode control [6–8], etc. All of the above methods have been successfully applied in UUV trajectory tracking research.

In addition to the above work, model predictive control has been widely used in UUV control owing to its own advantages of adding constraints. Model predictive control has the inherent advantage of dealing with the internal constraints of the machine directly [9–12]. It has a strong adaptability to the nonlinear underwater environment, and can solve

This project is supported by the National Natural Science Foundation of China (62033009) and the Creative Activity Plan for Science and Technology Commission of Shanghai (23550730300).

the problems of speed jump and thrust saturation well in the tracking process [13–15]. Gan et al. [16] proposed the MPC method based on quantum particle swarm optimization (QPSO), which adopts the latest optimization algorithm, to achieve the faster tracing with constraints. In reference [17], an observer was designed to improve the robustness of UUV by adding the influence of ocean currents. In reference [18], a hybrid control strategy based on MPC is applied to 7000m-Human Occupied Vehicle and its feasibility is verified, and Zhu et al. proposed further model predictive cascade control for resolving actuator saturation in human-occupied vehicle trajectory tracking. Yao et al. [20] proposed an improved model predictive control and designed different weight for comparison according to different depths in the fixed-depth control. It verifies the feasibility of the method and gives the influence of different size weight. The above research solved the thrust saturation, but failed to improve the UUV speed control value. In addition, fixed weight leads to a decrease in UUV adaptive ability. Its tracking accuracy, convergence speed and robustness all need to be further improved.

In this paper, inspired by adaptive weight control and constraint optimization of MPC, a hybrid control strategy QPSO-AWMPC which combines quantum particle swarm optimization (QPSO) and adaptive weight model predictive control (AWMPC) is proposed to deal with the tracking control problem of UUV. Then the SMC method was cascaded to realize UUV dynamic design. The overall strategy is comprised of two parts: the first part is the kinematics controller based on QPSO-AWMPC, which can get the desired speed according to the position information. The second part is the dynamics controller based on SMC, which can get the desired thrust according to the speed information.

The rest of the paper is organized as follows. In Section II, the control principle of MPC, the optimization algorithm of QPSO, the simplified model and thruster arrangement of UUV are introduced. In Section III, we present the hybrid

control strategy QPSO-AWMPC, and design the controller of kinematics and dynamics. In Section IV, the stability analysis of the controller is given. In Section V, to indicate the feasibility and effectiveness of the proposed method, simulation comparison and analysis are given. In Section VI, concluding remarks are made.

## II. Algorithm principle and UUV Simplified model

### A. Algorithm principle of model predictive control

For model predictive control, there are three key steps in the process, namely three basic principles, which are prediction model, rolling optimization and feedback correction, respectively corresponding to the prediction model, controller and corrector. The function of the prediction model is to predict the future output value. The controller identifies the characteristics of the system output. And the corrector is the feedback correction for the error.

During the control process, there is always an expected reference trajectory used as the optimization target. Taking time $k$ as the current time, the controller combines the current measured value and prediction model. To predict the output of the system in the future domain $[k, k + N_P]$ ($N_P$ is the predicted time domain), a series of control sequences in the control time domain $[k, k + N_C]$ ($N_C$ is the control time domain) is obtained by solving optimization problems satisfying objective functions and constraints.

The first element of the control sequence is taken as the actual control quantity of the controlled object. When it comes to the next time $k + 1$, the above process repeats, completing an optimization problem with constraints in a rolling manner, thereby achieving continuous control of the controlled object.

### B. QPSO algorithm

Quantum particle swarm optimization (QPSO) is a type of swarm intelligence optimization algorithm, which is developed from ant colony algorithm, particle swarm optimization algorithm, etc. The QPSO algorithm cancels the movement direction attribute of the particle. And the update of the particle position has nothing to do with the previous movement of the particle, increasing the randomness of the particle position. The biggest update is that the new quantity $mbest$ is introduced to represent the average value of $pbest$. The objective function is optimized by QPSO algorithm to obtain the control sequence. The pseudocode is shown in Table I.

The control increment $V$ is initialized. Then calculating the fitness functions value of each group and finding out the optimal group. Depending on the corresponding formula, the above control increment is updated, the fitness function value is calculated, the optimal control increment is found and compared with the previous increment. The specific implementation process of QPSO method. In this way, when the cycle number reaches the maximum, the optimal control increment is achieved.

TABLE I
PSEUDOCODE OF QPSO

**Algorithm QPSO**

**Input:** particle $i$ (N is the popsize of particle)
**Output:** $Gbest$
1: **for** each particle $i$
2:   Initialize velocity $V_i$ and position $X_i$
3:   Evaluate particle $i$ and set $Pbest_i = X_i$
4: **end for**
5: $Gbest = \min(Pbest)$
6: **while** not stop
7:   **for** $i = 1$ to $N$
8:     Update the velocity and position
9:     Evaluate particle $i$
10:     **if** fitness$(X_i) <$ fitness$(Pbest_i)$
11:       $Pbest_i = X_i$
12:     **if** fitness$(Pbest_i) <$ fitness$(Gbest_i)$
13:       $Gbest_i = Pbest_i$
14:   **end for**
15: **end while**
16: print $Gbest$

### C. UUV dynamics simplified model

UUV moves in space with six degrees of freedom under water. Its coordinate system is divided into inertial frame and body-fixed frame, as shown in Fig. 1.

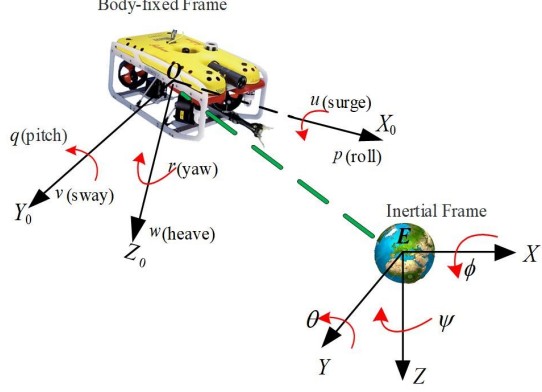

Fig. 1. UUV coordinate system.

The motion in the body-fixed frame can be expressed by the linear motion along three axes and the rotation around three axes, and its nonlinear dynamic motion equation is as follows:

$$M\dot{V} + C(v)V + D(v)V + g(\eta) = \tau \qquad (1)$$

where $M$ is the inertia matrix, $C(v)$ is the Coriolis term and the centrifugal term, $D(v)$ is hydrodynamic loss term, $g(\eta)$ is the resultant force term of gravity and buoyancy, $\tau$ is the input for propeller control, $V = \begin{bmatrix} u & v & w & p & q & r \end{bmatrix}^T$. In practical applications, pitch and roll movements are rare, so this paper studied the relatively frequent motions, namely surge, sway, heave and yaw. The reference state is:

$$\eta_d(t) = \begin{bmatrix} x_d(t) & y_d(t) & z_d(t) & \psi_d(t) \end{bmatrix}^T \qquad (2)$$

The real-time state is:

$$\eta(t) = \begin{bmatrix} x(t) & y(t) & z(t) & \psi(t) \end{bmatrix}^T \qquad (3)$$

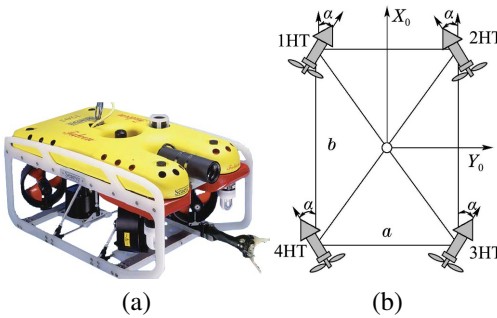

Fig. 2. Falcon and its thruster layout.

The errors of $x, y, z, \psi$ are respectively $e_x, e_y, e_z, e_\psi$. The state vector in the inertial frame is expressed as $\eta = \begin{bmatrix} x & y & z & \psi \end{bmatrix}^T$. The speed vector in body-fixed frame is expressed as $v = \begin{bmatrix} u & v & w & r \end{bmatrix}^T$.

The correlation is:

$$\dot{\eta} = J(\eta)v \tag{4}$$

That is:

$$\begin{bmatrix} \dot{x} \\ \dot{y} \\ \dot{z} \\ \dot{\psi} \end{bmatrix} = \begin{bmatrix} \cos\psi & -\sin\psi & 0 & 0 \\ \sin\psi & \cos\psi & 0 & 0 \\ 0 & 0 & 1 & 0 \\ 0 & 0 & 0 & 1 \end{bmatrix} \begin{bmatrix} u \\ v \\ w \\ r \end{bmatrix} \tag{5}$$

where $J(\eta)$ is the Jacobian matrix, which is generally used as the transformation matrix from the body-fixed frame to the inertial frame.

### D. Thrusters arrangement of Falcon and thrust normalization

To check the effectiveness of the proposed algorithm, we used the data of Falcon vehicle for simulation, so the thruster system of Falcon is analyzed. The structure and horizontal thruster layout of Falcon are shown in Fig. 2.

Falcon unmanned underwater vehicle contains 5 thrusters, 4 thrusters in the horizontal plane and 1 thruster in the vertical plane. It can carry out 4-dof movements: surge, sway, yaw and heave. $a$ and $b$ are width and length of Falcon, $\alpha$ is angle between the thruster and the $X_0$ axis of the body-fixed frame. The specific data are as follows: a = 0.6 m, b = 1 m, and $\alpha = 36°$.

The hydrodynamic parameters of Falcon can refer to [16]. The simplified dynamic model used in the simulation is as follows:

$$\begin{cases} (m + X_{\dot{u}})\,\dot{u} + X_u u + X_{uu}u|u| = \tau_X \\ (m + Y_{\dot{v}})\,\dot{v} + Y_v v + Y_{vv}v|v| = \tau_Y \\ (m + Z_{\dot{w}})\,\dot{w} + Z_w w + Z_{wv}w|w| = \tau_Z \\ (I_z + N_{\dot{r}})\,\dot{r} + N_r r + N_{rr}r|r| = \tau_\psi \end{cases} \tag{6}$$

According to the propeller layout of Fig. 2 and the principle of dynamic, the resultant force and resultant moment generated by Falcon on each degree of freedom can be easily calculated:

$$\begin{bmatrix} \tau_X \\ \tau_Y \\ \tau_Z \\ \tau_\Psi \end{bmatrix} = \begin{bmatrix} \cos\alpha & \cos\alpha & \cos\alpha & \cos\alpha & 0 \\ \sin\alpha & -\sin\alpha & \sin\alpha & -\sin\alpha & 0 \\ 0 & 0 & 0 & 0 & 1 \\ A & -A & -A & A & 0 \end{bmatrix} \begin{bmatrix} T_1 \\ T_2 \\ T_3 \\ T_4 \\ T_5 \end{bmatrix} \tag{7}$$

where $\begin{bmatrix} \tau_X & \tau_Y & \tau_Z & \tau_N \end{bmatrix}^T$ are the resultant force and moment on the four degrees of freedom, and $\begin{bmatrix} T_1 & T_2 & T_3 & T_4 & T_5 \end{bmatrix}^T$ are thrust on each propeller. $A = (b/2) \cdot \sin\alpha + (a/2) \cdot \cos\alpha$. The thrust can be normalized and obtained:

$$\begin{bmatrix} \bar{\tau}_X \\ \bar{\tau}_Y \\ \bar{\tau}_Z \\ \bar{\tau}_N \end{bmatrix} = \begin{bmatrix} 0.25 & 0.25 & 0.25 & 0.25 & 0 \\ 0.25 & -0.25 & 0.25 & -0.25 & 0 \\ 0 & 0 & 0 & 0 & 1 \\ 0.25 & -0.25 & -0.25 & 0.25 & 0 \end{bmatrix} \begin{bmatrix} \bar{T}_1 \\ \bar{T}_2 \\ \bar{T}_3 \\ \bar{T}_4 \\ \bar{T}_5 \end{bmatrix} \tag{8}$$

That is:

$$\begin{bmatrix} \bar{\tau}_X \\ \bar{\tau}_Y \\ \bar{\tau}_Z \\ \bar{\tau}_N \end{bmatrix} = \bar{B} \begin{bmatrix} \bar{T}_1 \\ \bar{T}_2 \\ \bar{T}_3 \\ \bar{T}_4 \\ \bar{T}_5 \end{bmatrix} \Leftrightarrow \bar{\tau} = \bar{B} \cdot \bar{T} \tag{9}$$

$$\bar{T} = \bar{B}^{-1} \cdot \bar{\tau} \tag{10}$$

where $\bar{\tau}$ is the normalized form of $\tau$, and $\bar{T}$ is the normalized form of $T$, $-1 \leq \bar{T}_i \leq 1, i = 1, 2, 3, 4, 5$.

## III. CONTROL STRATEGY AND SYSTEM DESIGN

### A. Objective function and adaptive weight design

The objective function is designed as:

$$J(k) = \sum_{i=1}^{N_p} \|\eta(k + i \mid t) - \eta_{ref}(k + i \mid t)\|_{Q_\eta}^2 \\ + \sum_{i=0}^{N_c - 1} \|\Delta V(k + i \mid t)\|_{R_r}^2 \tag{11}$$

The first term is the path tracking error penalty term, whose physical meaning is the ability to track the expected path. $Q_\eta$ is the state weight matrix. In this design, $Q_\eta$ has a fixed value. The second is the control input penalty term, which suppresses the excessively drastic control increment to ensure that there will be no big jump at the control input port, making the input performance more stable. $R_v$ is the control input weight matrix. In this design, $R_v$ takes variable values.

In most studies, the weight matrix is fixed, like in references [11], [16]. Considering the actual track tracking process, the controller has different requirements for different stages of UUV operation. There is a large error in the initial state of tracking, while the controller expects the UUV to approach the reference trajectory at a higher speed. The tracking error is small in the late stage, while the controller expects the UUV to operate at a more stable state and speed. To meet this demand,

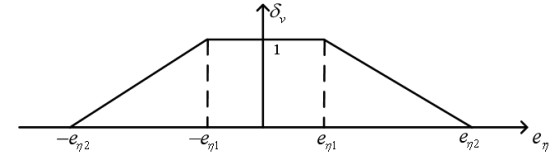

Fig. 3. Diagram of relationship between $\delta_v$ and $e_\eta(t)$.

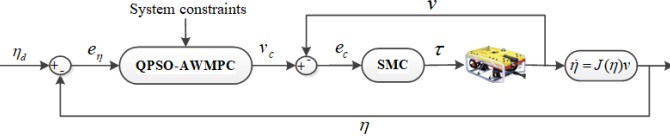

Fig. 4. Control system flowchart.

the MPC method based on adaptive weight is designed in this paper. The control weight matrix $R_v$ is designed to be variable, which makes UUV converge at a relatively higher speed and operate in a more stable state. UUVs specific constraint must be met in the process, and the MPC method handles this well.

This paper introduces a variable parameter $\delta_v$. Weight $R_v$ is expressed as:

$$R_v = \delta_v I_{m \times m} \tag{12}$$

$I_{m \times m}$ is the $m$-dimensional identity matrix, and $m$ is the number of controlled-quantity.

The relationship between $\delta_v$ and state error $e_\eta(t)$ is shown in Fig. 3.

Here, $e_\eta(t) = \begin{bmatrix} x - x_d & y - y_d & z - z_d & \psi - \psi_d \end{bmatrix}$, $e_\eta = \|e_\eta(t)\|$, $e_{\eta1}, e_{\eta2}$ is the state error of different sizes, and $e_{\eta1} < e_{\eta2}$. When $\|e_\eta(t)\| \geq e_{\eta2}$, the value of $\delta_v$ approaches the minimum value of 0 , so as to give a large control increment and improve the tracking speed. When $\|e_\eta(t)\| \leq e_{\eta1}$, the value of $\delta_v$ approaches the maximum value of 1 , so as to suppress large fluctuations of control variables and improve stability. When $e_{\eta1} < \|e_\eta(t)\| < e_{\eta2}$, there is a law of change. Where 0 and 1 are theoretical limits, the actual value is close to 0 and 1. The law of global change is expressed as follows:

$$\begin{cases} \delta = 0 & (e_\eta \geq \|e_{\eta2}\|) \\ \delta = \frac{e_{\eta2} - \|e_\eta\|}{e_{\eta2} - e_{\eta1}} & (e_{\eta1} < \|e_\eta(t)\| < e_{\eta2}) \\ \delta = 1 & (e_\eta \leq \|e_{\eta1}\|) \end{cases} \tag{13}$$

B. Design of kinematics controller

The entire system design flow chart is shown in Fig. 4, and it is subdivided into kinematics controller design as the outer loop and dynamics controller design as the inner loop.

Here, $\eta_d$ is the reference state, $e_\eta$ is the state error, $v_c$ is the expected speed, $\eta$ is the real-time state, $v$ is the real-time speed. The kinematics controller is comprised of error model, system constraint and objective function. The state error in the inertial frame is optimized, while the kinematics controller is constructed to obtain the desired speed $v_c$.

*1) Error model:* The optimal object for kinematic tracking is the state variable error. The UUV tracks the reference trajectory on the status by narrowing the error. Reference state and the actual state are $\eta_d(t), \eta(t)$ respectively, So the state error is defined as:

$$e_\eta(t) = \eta_d(t) - \eta(t) \tag{14}$$

When $t$ tends to infinity, the state error of UUV tends to 0, that is:

$$\lim_{t \to \infty} e_\eta(t) = 0 \tag{15}$$

With the speed as the input of the system and the state as the output of the system, the real-time system can be expressed as follows:

$$\dot{\eta} = f(\eta, v) \tag{16}$$

Each point needs to satisfy the above equation, then the reference trajectory can be expressed as:

$$\dot{\eta}_d = f(\eta_d, v_d) \tag{17}$$

In general, Taylor expansion method is employed to error analysis. Carrying out Taylor expansion of equation (16) at the reference point and ignoring higher-order terms:

$$\dot{\eta} = f(\eta_d, v_d) + \left.\frac{\partial f(\eta, v)}{\partial \eta}\right|_{\substack{\eta = \eta_d \\ v = v_d}} (\eta - \eta_d) + \left.\frac{\partial f(\eta, v)}{\partial v}\right|_{\substack{\eta = \eta_d \\ v = v_d}} (v - v_d) \tag{18}$$

Equation (17) minus equation (16) is equal to:

$$\dot{\tilde{\eta}} = A(t) \begin{bmatrix} x - x_d \\ y - y_d \\ z - z_d \\ \psi - \psi_d \end{bmatrix} + B(t) \begin{bmatrix} u - u_d \\ v - v_d \\ w - w_d \\ r - r_d \end{bmatrix} \tag{19}$$

where:

$$A(t) = \begin{bmatrix} 0 & 0 & 0 & -u\sin\psi - v\sin\psi \\ 0 & 0 & 0 & u\cos\psi - v\cos\psi \\ 0 & 0 & 0 & 0 \\ 0 & 0 & 0 & 0 \end{bmatrix}$$

$$B(t) = \begin{bmatrix} \cos\psi & -\sin\psi & 0 & 0 \\ \sin\psi & \cos\psi & 0 & 0 \\ 0 & 0 & 1 & 0 \\ 0 & 0 & 0 & 1 \end{bmatrix}$$

Then, the above equation can be expressed as:

$$\dot{\tilde{\eta}} = A(t)\tilde{\eta}(k) + B(t)\tilde{v}(k) \tag{20}$$

In fact, the controller design is a discrete sampling, so the first-order difference quotient method is adopted to discretize the continuous equation:

$$A_{k,t} = I + T \cdot A(t) \tag{21}$$

$$B_{k,t} = T \cdot B(t) \tag{22}$$

$T$ is the sampling time, that is:

$$A_{k,t} = \begin{bmatrix} 1 & 0 & 0 & (-u\sin\psi - v\sin\psi)T \\ 0 & 1 & 0 & (u\cos\psi - v\cos\psi)T \\ 0 & 0 & 1 & 0 \\ 0 & 0 & 0 & 1 \end{bmatrix}$$

$$B_{k,t} = \begin{bmatrix} T\cos\psi & -T\sin\psi & 0 & 0 \\ T\sin\psi & T\cos\psi & 0 & 0 \\ 0 & 0 & T & 0 \\ 0 & 0 & 0 & T \end{bmatrix}$$

Then, equation (20) can be expressed as:

$$\tilde{\eta}(k+1) = A_{k,t}\tilde{\eta}(k) + B_{k,t}\tilde{v}(k) \tag{23}$$

*2) Solution of control:* We change equation (23) to the following form and get an augmented state-space model:

$$\xi(k \mid t) = \begin{bmatrix} \tilde{\eta}(k \mid t) \\ \tilde{v}(k-1 \mid t) \end{bmatrix} \tag{24}$$

The augmented model can be written in the following matrix form:

$$\xi(k+1 \mid t) = \tilde{A}_{k,t}\xi(k \mid t) + \tilde{B}_{k,t}\Delta v(k \mid t) \tag{25}$$

$$\eta(k \mid t) = \tilde{G}_{k,t}\xi(k \mid t) \tag{26}$$

where, $\tilde{A}_{k,t} = \begin{bmatrix} A_{k,t} & B_{k,t} \\ 0m \times n & I_m \end{bmatrix}$, $\tilde{B}_{k,t} = \begin{bmatrix} B_{k,t} \\ I_m \end{bmatrix}$, $\tilde{G}_{k,t} = \begin{bmatrix} G_{k,t} & 0 \end{bmatrix}$, $m = 4$, $n = 4$.

The predicted outputs sequence is in the finite prediction layer $N_P$ and the control input sequence is in the finite control layer $N_C$:

The output sequence is:

$$\gamma = \begin{bmatrix} \eta(k+1 \mid k) & \eta(k+2 \mid k) & \cdots & \eta(k+N_p \mid k) \end{bmatrix}^T \tag{27}$$

The control input sequence is:

$$\Delta V = \begin{bmatrix} \Delta v(k) & \Delta v(k+1) & \cdots & \Delta v(k+N_c-1) \end{bmatrix}^T \tag{28}$$

We combine equations of (25), (26), (27) and (28) to obtain the system prediction output:

$$\gamma = \Psi_t \xi(t \mid t) + \Theta_t \Delta V(t) \tag{29}$$

where

$$\Psi_t = \begin{bmatrix} \tilde{G}_{t,t}\tilde{A}_{t,t} \\ \cdots \\ \tilde{G}_{t,t}\tilde{A}_{t,t}^{N_p} \end{bmatrix}$$

$$\Theta_t = \begin{bmatrix} \tilde{G}_{t,t}\tilde{B}_{t,t} & 0 & 0 & 0 \\ \tilde{G}_{t,t}\tilde{A}_{t,t}\tilde{B}_{t,t} & \tilde{G}_{t,t}\tilde{B}_{t,t} & 0 & 0 \\ \tilde{G}_{t,t}\tilde{A}_{t,t}^{N}\tilde{B}_{t,t} & \tilde{G}_{t,t}\tilde{A}_{t,t}^{N_t-1}\tilde{B}_{t,t} & \vdots & \tilde{G}_{t,t}\tilde{A}_{t,t}\tilde{B}_{t,t} \\ \vdots & \vdots & \ddots & \vdots \\ \tilde{G}_{t,t}\tilde{A}_{t,t}^{N_p-1}\tilde{B}_{t,t} & \tilde{G}_{t,t}\tilde{A}_{t,t}^{N_R-2}\tilde{B}_{t,t} & \cdots & \tilde{G}_{t,t}\tilde{A}_{p}^{N_p-N_c-1}\tilde{B}_{t,t} \end{bmatrix}$$

Substitute equation (29) into the target function (11):

$$J(k) = \Delta V(t)H_t\Delta V(t)^T + G_t\Delta V(t)^T \tag{30}$$

where $H_t = \begin{bmatrix} \Theta_t^T Q\Theta_t + R & 0 \end{bmatrix}$, $G_t = \begin{bmatrix} 2E(t)^T Q\Theta_t & 0 \end{bmatrix}$.

The simulation constraints are set as follows. The control quantity constraint:

$$-v_{\max} \leq v(k+n) \leq v_{\max}, n = 0, 1, \cdots N_c - 1 \tag{31}$$

The control increment constraint:

$$-v_{\max} \leq \Delta v(k+n) \leq v_{\max}, n = 0, 1, \cdots N_c - 1 \tag{32}$$

Based on lots of data, we set $v_{\max} = 2$ m/s. The terminal item of prediction output $\gamma$ is as follows, which is set as 0.

$$\eta(k+N_p \mid k) = \tilde{G}_{t,t}\tilde{A}^{N_P}\xi(t,t)$$
$$+ \begin{bmatrix} \tilde{G}_{t,t}\tilde{A}_{t,t}^{N_P-1}\tilde{B}_{t,t} & \cdots & \tilde{G}_{t,t}\tilde{A}_{t,t}^{N_P-N_C-1}\tilde{B}_{t,t} \end{bmatrix}\Delta V = 0 \tag{33}$$

In conclusion, the kinematic trajectory tracking problem can be transformed into the following optimization problem.

$$\min_{\Delta V} J(k) = \Delta V(t)H_t\Delta V(t)^T + G_t\Delta V(t)^T \tag{34}$$
$$\text{s.t.} \quad Eq(31) - Eq(33)$$

The control sequence for a period of time in the future can be obtained by solving this equation with QPSO algorithm. Then, its optimal solution can be obtained and the first increment of $\Delta V^*$ is applied as the control input:

$$\Delta V^* = \begin{bmatrix} 1 & 0 & \cdots & 0 \end{bmatrix}\begin{bmatrix} \Delta v_t & \Delta v_{t+1} & \cdots & \Delta v_{t+N_C-1} \end{bmatrix}^T \tag{35}$$

*C. Design of dynamic controller*

Kinematics system output is the desired speed. In order to get thrust to reach the desired speed, a dynamic controller needs to be built. The design of the controller adopts the sliding mode control method.

Due to the chattering caused by switching terms in traditional SMC, adaptive continuous switching terms are designed in this paper to avoid intermittent jumping.

$v$ and $v_c$ are the real-time speed and desired speed of UUV respectively. Then the speed tracking error is defined as:

$$e_c = v_c - v \tag{36}$$

Sliding mode control includes the design of sliding mode surface and control law. Firstly, the design of sliding mode surface is as follows:

$$s = \dot{e}_c + 2\Lambda e_c + \Lambda^2 \int e_c dt \tag{37}$$

We take the derivative of $S$ and set it to 0:

$$\dot{s} = \ddot{e}_c + 2\Lambda\dot{e}_c + \Lambda^2 e_c = \ddot{e}_c + 2\Lambda(\dot{v}_c - \dot{v}) + \Lambda^2 e_c = 0 \tag{38}$$

The known kinetic equation is:

$$M\dot{v} + C(v)V + D(v)V + g(\eta) = \tau \tag{39}$$
$$\dot{v} = M^{-1}(\tau - (C(v)V + D(v)V + g(\eta))) \tag{40}$$

We put equation (40) into equation (38) to get:

$$\ddot{e} + 2\Lambda\left(\dot{v}_c - M^{-1}(\tau - (C(v) + D(v) + g(\mu)))\right) + \Lambda^2 e = 0 \tag{41}$$

Because the UUV dynamic model is not completely known, the dynamic equation is equivalent to the estimated term and the unknown term, so:

$$\tau = \hat{\tau} + \tilde{\tau} \tag{42}$$

where, $\tilde{\tau} = \tilde{M}q + \tilde{C}q + \tilde{D}q + \tilde{g} + \tau_w$; $\hat{\tau} = \hat{M}q + \hat{C}q + \hat{D}q + \hat{g}$; $M, C, D,$ g are estimate term of $\hat{M}, \hat{C}, \hat{D}, \hat{g}$, $M, C, D, g$ are unknown term of $\tilde{M}, \tilde{C}, \tilde{D}, \tilde{g}$, $\tau_w$ is unknown disturbance. The control law is designed as follows:

$$\tau_{eq} = \hat{M}\left(\dot{v}_c + \frac{\ddot{e}_c}{2\Lambda} + \frac{\Lambda}{2}e_c\right) + \hat{C}q + \hat{D}q + \hat{g} \tag{43}$$

Because UUV runs slowly underwater, the unknown term can be considered to be very small and bounded. In order to ensure the stability of the sliding mode surface, the adaptive term is designed and added into the control law.

$$\tau_{ad} = \Gamma s + \left(K + \frac{\hat{C}}{2\Lambda}\right)s \tag{44}$$

The complete control law is:

$$\tau = \tau_{eq} + \Gamma s + \left(K + \frac{\hat{C}}{2\Lambda}\right)s \tag{45}$$

## IV. SIMULATION AND RESULT ANALYSIS

This experiment compared the improved QPSO-AWMPC method with the fixed-weighted QPSO-MPC and backstepping methods. The experiment was split into three-dimensional straight line and spiral curve simulation, respectively to verify the effect of speed jump or thrust saturation problems and convergence speed improvement.

Related parameters: the sampling period was set as $T_s = 0.1$ s, while sampling time is $T = 200$ s and $T = 500$ s. Backstepping parameters: $\Lambda = 3, K = 60, \Gamma = 100$. AWMPC parameters: population size popsize $= 40$, maximal iterations $MAXITER = 50$, predictive domain $N_p = 10$, control domain $N_c = 10$; dynamics related parameters: $\Lambda = 3, K = 60$, $\Gamma = 100$. Due to the insignificant changes of all parameters after a period of simulation, the intercept time of some simulation diagrams is less than the total simulation time.

An important point is explained here in advance. If backstepping method does not take account of the actual constraints, the reference trajectory will be tracked quickly. In order to demonstrate the actual situation, we will limit the maximum thrust of the simulation.

1) The control law of backstepping is:

$$\mathbf{v}_c = \begin{bmatrix} u_c \\ v_c \\ w_c \\ r_c \end{bmatrix} = \begin{bmatrix} k\left(V_x \cos\psi + V_y \sin\psi\right) + \left(u_d \cos e_\psi - v_d \sin e_\psi\right) \\ k\left(-V_x \sin\psi + V_y \cos\psi\right) + \left(u_d \sin e_\psi + v_d \cos e_\psi\right) \\ w_d + k_z V_z \\ r_d + k_\psi V_\psi \end{bmatrix} \tag{46}$$

where $k, k_z, k_\psi$ is the positive constant.

2) The weight in method QPSO-MPC is fixed value 0.5.

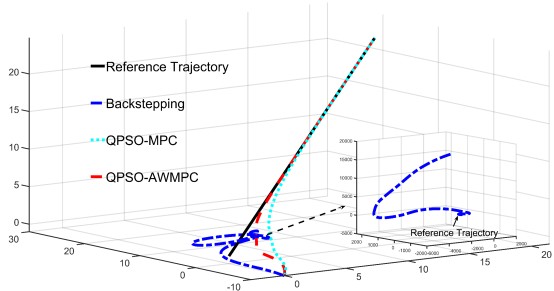

Fig. 5. Comparison diagram of straight line tracking.

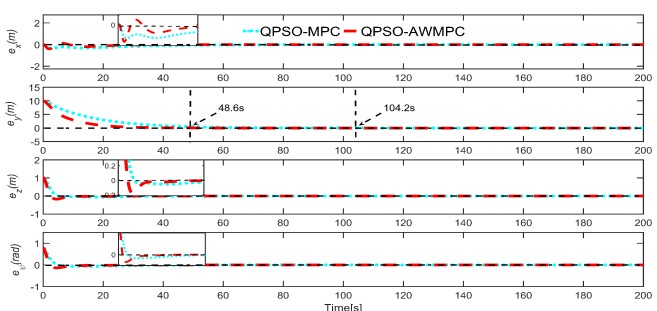

Fig. 6. Comparison diagram of tracking error changes.

### A. Straight line tracking

By setting the initial state of UUV as $(0, -10, -1, 1)$, the reference trajectory is: $\eta_d(t) = \begin{cases} x_d = 0.15 * t \\ y_d = 0.15 * t \\ z_d = 0.15 * t \\ psid = \pi/4 \end{cases}$ The simulation results are shown in Figs. 5-8.

The tracking result is given in Fig. 5. The blue dotted line represents the tracking trajectory generated by backstepping method. It can be seen that its track trajectory is divergent in finite space and cannot be converge, which is caused by the constraint of its thrust. The other two methods can track the reference trajectory stably. And the proposed QPSO-AWMPC tracking effect is better. Because backstepping method cannot track the reference trajectory, we will not talk about it in error

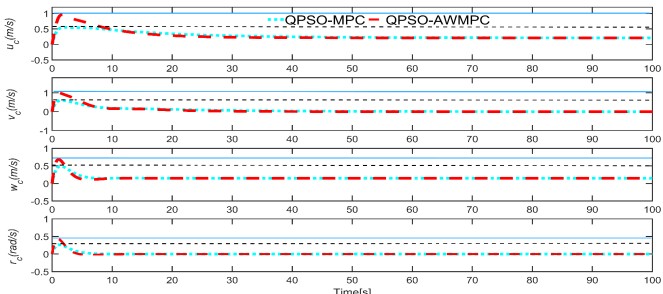

Fig. 7. Comparison diagram of speed changes.

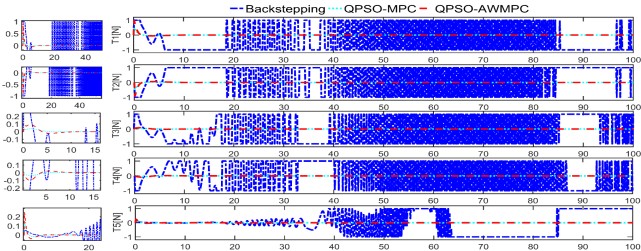

Fig. 8.    Comparison diagram of normalized thrusts.

TABLE II
CONTRAST OF ERROR ACCURACY IN 20S

| Method | X | Y | Z | $\psi$ |
|---|---|---|---|---|
| QPSO-MPC | -0.277 | 5.493 | -0.038 | -0.065 |
| QPSO-AWMPC | -0.082 | 2.956 | -0.025 | -0.043 |

TABLE IV
THE COMPARISON OF MAXIMUM SPEED

| Method | $u_c$ (m/s) | $v_c$ (m/s) | $w_c$ (m/s) | $r_c$ (m/s) |
|---|---|---|---|---|
| QPSO-MPC | 0.5481 | 0.5941 | 0.4885 | 0.2647 |
| QPSO-AWMPC | 0.9621 | 0.9954 | 0.6830 | 0.4322 |

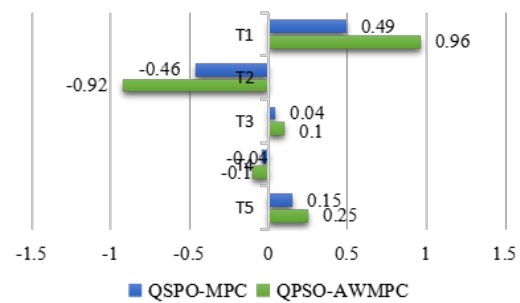

Fig. 9.    Maximal normalized thrust.

and speed analysis.

Fig. 6 shows the error variation in the tracking process. Taking the X direction as an example, we compare the tracking accuracy of the two methods with a fixed horizontal axis and find that the error of QPSO-AWMPC is always smaller than that of QPSO-MPC at the same time. Selecting 20s as an example to compare the error accuracy and the results are shown in Table II. We fixed the vertical axis to compare the taken time when the error was about 0, and found that QPSO-AWMPC method could converge to the vicinity of 0 faster. Taking the $Y$ direction as an example, final convergence time of QPSO-AWMPC and QPSO-MPC is 48.6s and 104.2s respectively, and the convergence time is shown in Table III. This method saves a lot of tracking time and has a higher tracking accuracy.

The speed changes can be observed in Fig. 7. The black dot line marks the maximum speed that the QPSO-MPC method can achieve, and the blue solid line marks the maximum speed that the QPSO-AWMPC method can achieve. In large error phase, the speed of latter is always greater in four directions at the same time. In small error phase, such as 15-30s, the latter speed is lower than the former, which is caused by the latter larger control weight. Due to the large error of X and Y axis directions, the speed difference in these two directions is more obvious in the tracking initial stage: in the Y direction, the maximum speed of QPSO-AWMPC reaches about 0.9954 m/s, while the maximum speed of QPSO-MPC is about 0.5941 m/s. And the maximum speed of two methods satisfies the constraint. In the tracking stable phase, the two speeds are nearly equal. This speed change shows that QPSO-AWMPC converges faster in the early stage of the tracking process.

TABLE III
TACKING TIME WHEN THE ERROR IS 0

| Method | Backstepping | QPSO-AWMPC | QPSO-MPC |
|---|---|---|---|
| Time | $+\infty$ | 48.6s | 104.2s |

Maximum speed value of UUV is shown in the Table IV.

Fig. 8 shows the thrust change after normalization. After cutting off the over-limit thrust of backstepping method, the track diverges and the subsequent thrust changes greatly. The thrust of both QPSO-MPC and QPSO-AWMPC is within the constraint range. In the initial stage, the thrust of QPSO-AWMPC is larger than that of QPSO-MPC, while the thrust of tracking stable phase is almost equal. The thrust magnitude reflects the speed magnitude, so this result proves that QPSO-AWMPC tracking converges faster again. The maximum thrust value is illustrated in Fig. 9.

In conclusion, backstepping cannot track the reference trajectory under the condition of limited thrust. QPSO-MPC can track the reference trajectory well, but its convergence speed and tracking accuracy are far less than QPSO-AWMPC.

### B. Spiral curve tracking with constraints

By setting the initial state of UUV as $(-10, 10, 0, 1)$, the reference trajectory is: $\eta_d(t) = \begin{cases} x_d = 25 * \sin(0.02 * t) \\ y_d = -25 * \cos(0.02 * t) \\ z_d = 0.1 * t \\ \text{psid} = 0.02 * t \end{cases}$,

The simulation results are shown in Figs. 10-13.

Fig. 10 shows the tracking results of the three methods in the presence of thrust constraints. The backstepping method cannot effectively track the reference trajectory under the thrust constraint. However, both QPSO-AWMPC and QPSO-MPC can track the reference trajectory. QPSO-AWMPC has a better tracking result than QPSO-MPC.

Fig. 11 shows the normalized thrust change. In order to respond to the actual situation, the thrust generated by backstepping is limited to [-1, 1]. So thrust constraint makes it can't achieve trajectory tracking process. Both QPSO-AWMPC and QPSO-MPC are restricted within the constraint, and the former has bigger thrust.

From the error changes in the four directions in Fig. 12, it can be seen that the error reduction speed of QPSO-AWMPC

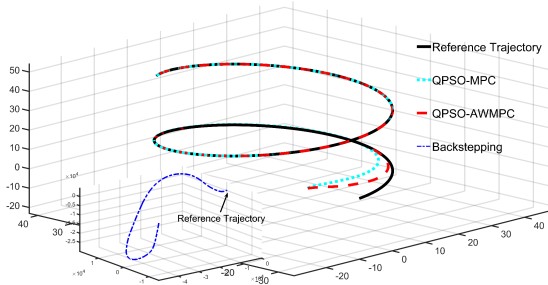

Fig. 10.    Comparison diagram of straight line tracking.

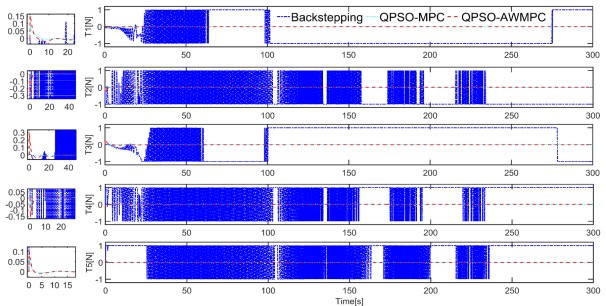

Fig. 11.    Comparison diagram of tracking error changes.

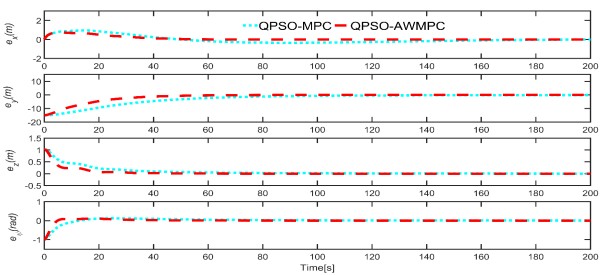

Fig. 12.    Comparison diagram of speed changes.

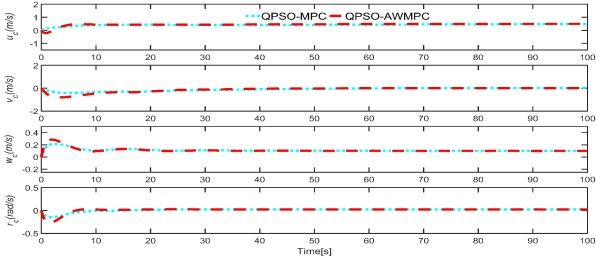

Fig. 13.    Comparison diagram of normalized thrusts.

TABLE V
CONTRAST OF ERROR ACCURACY IN 20S

| Method | X | Y | Z | $\psi$ |
|---|---|---|---|---|
| QPSO-MPC | 0.855 | -9.389 | 0.221 | 0.109 |
| QPSO-AWMPC | 0.503 | -4.584 | 0.066 | 0.098 |

TABLE VI
COMPARISON OF CONVERGENCE TIME

| Method | Backstepping | QPSO-AWMPC | QPSO-MPC |
|---|---|---|---|
| Time | $+\infty$ | 50.3s | 100.9s |

method is greater and the tracking accuracy is higher. In four directions, at the same time, the error brought by QPSO-AWMPC is always smaller than that of QPSO-MPC, and the former fluctuation is slight, so the final error is closer to 0. The error accuracy in 20s is shown in Table V. The convergence time can also be obtained from the error variation and is shown in Table VI.

Fig. 13 shows the tracking real-time speed, and the initial real-time speed which represents the tracking convergence speed. As can be seen from the figure, speed of QPSO-AWMPC in all four directions is greater than that of QPSO-MPC method, while the latter speed is basically same.

### C. Robustness analysis

To verify tracking stability, we added disturbance at the 320s to simulate thruster failure. Observing trajectory trends and the result was shown in Fig. 14. In the process of UUV tracking, we alter its speed. At this time, the UUV cannot reach the expected speed, so the tracking trajectory will deviate from the reference trajectory. From the degree of trajectory deviation, it can be seen that the tracking trajectory of QPSO-AWMPC is closer to the reference trajectory, with smaller relative error and higher accuracy. When the error reaches the maximum point, the trajectory descent gradient of QPSO-AWMPC is grater, that is to say, its convergence speed is faster than that of QPSO-MPC. In conclusion, the improved method had better robustness, accuracy and speed.

Through simulation experiments of straight line and spiral curve tracking, on the basis of solving the problem of thrust saturation, the improved method can track the reference trajectory quickly with higher accuracy. The convergence time is about 50-60s less than QPSO-MPC method. Simulation experiment of robustness analysis shows that the robustness of UUV tracking process in the later stage is better, and the recovery speed is faster. The simulation results verify the effectiveness of the QPSO-AWMPC method.

## V. CONCLUSION

In this paper, an improved MPC method based on QPSO is designed to improve the convergence speed and accuracy of UUV trajectory tracking. In this method, the control weight is a function of state error. In the initial stage, if the error is large, the weight is small and the tracking speed is greater. In the later stage, if the error is small, the weight is large and the

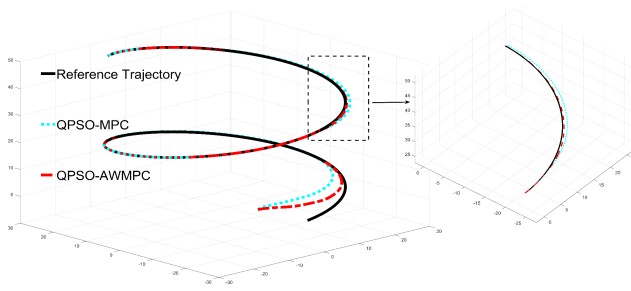

Fig. 14.    Comparison diagram of speed changes.

operation robustness is greater. Finally, the effectiveness of this method is verified by MATLAB simulation. This paper studied the convergence speed, accuracy and robustness deeply, and the fault tolerant control of tracking will be further discussed in the future work.

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
