# OpenReview forum: "An Adaptive Weight Model Predictive Control Algorithm to Trajectory Tracking Control of UUV"
_IEEE.org/ICIST/2024/Conference — IEEE ICIST 2024 Conference Submission_

### Official Review · Reviewer_PHD1 · 2024-08-22
**The work presented in this paper is significant as it addresses key challenges in UUV trajectory tracking, such as handling non-linearities, internal constraints, and improving convergence speed and accuracy.**

**Rating:** 8
**Confidence:** 4

**Review:**

The work presented in this paper is significant as it addresses key challenges in UUV trajectory tracking, such as handling non-linearities, internal constraints, and improving convergence speed and accuracy. The integration of QPSO with an adaptive weight mechanism provides a novel solution that adapts to different stages of UUV operation, making it more efficient and reliable in real-world applications. The simulation results further confirm the superiority of the proposed method over traditional approaches. However, future work could focus on experimental validation and exploring other optimization techniques to further enhance the control strategy. Below is a list of comments that should be taken into account further when revising the paper.
1.	Improper use of articles or long sentence structures with separators might make it difficult to follow the paper. For example, the use of ‘the’ in the paper and the presentation of subordinate clauses. Please check the full text again and modify the grammar problems.
2.	Please highlight the contributions of the paper.
3.	Please add the necessary comments for Figures.
4.	Meanwhile, please elaborate on the future plans.

---

### Official Review · Reviewer_Fmt9 · 2024-08-27
**In this paper, an adaptive weight model predictive control based on quantum particle swarm optimization (QPSO-AWMPC) is presented.**

**Rating:** 7
**Confidence:** 3

**Review:**

a There are some grammatical mistakes and typos. Please examine the full text further and revise them.
b The references should be updated. Some closely related and new references should be added to show to further explain the novelty and innovation of the work.
c What are the significant differences between this study and previous studies? The author needs more explicit emphasis.

---

### Official Review · Reviewer_BxFn · 2024-08-28
**An Adaptive Weight Model Predictive Control Algorithm to Trajectory Tracking Control of UUV**

**Rating:** 7
**Confidence:** 2

**Review:**

In this paper, an adaptive weight model predictive control based on quantum particle swarm optimization (QPSO-AWMPC) is presented.
a There are some grammatical mistakes and typos. Please examine the full text further and revise them.
b The references should be updated. Some closely related and new references should be added to show to further explain the novelty and innovation of the work.

---

### Decision · Program_Chairs · 2024-09-06

Accept (Oral)